

# Modelling the genesis of equatorial podzols: age and implications on carbon fluxes

Cédric Doupoux[1], Patricia Merdy[1], Célia Régina Montes[2], Naoise Nunan[3], Adolpho José Melfi[4], Osvaldo José Ribeiro Pereira[2], Yves Lucas[1]

[1] Université de Toulon, PROTEE Laboratory, EA 3819, CS 60584, 83041 Toulon Cedex 9, France
[2] University of São Paulo, NUPEGEL, CENA, Av. Centenário, 303, CEP 13416-903 Piracicaba, SP, Brazil
[3] CNRS, iEES Paris, 78850 Thiverval-Grignon, France
[4] University of São Paulo, IEE, ESALQ, São Paulo, SP, Brazil

*Correspondence to*: Cédric Doupoux (cedric.doupoux@gmail.com)

**Keywords:** Podzol, modelling, carbon storage, Amazonia

**Abstract.** Amazonian podzols store huge amounts of carbon and play a key role in transferring organic matter to the Amazon river. In order to better understand their C dynamics, we modelled the formation of representative Amazonian podzol profiles by constraining both total carbon and radiocarbon. We determined the relationships between total carbon and radiocarbon in organic C pools numerically by setting constant C and 14C inputs over time. The model was an effective tool for determining the order of magnitude of the carbon fluxes and the time of genesis of the main carbon-containing horizons, i.e. the topsoil and deep Bh. We performed retro calculations to take in account the bomb carbon in the young topsoil horizons ($^{14}$C age from 62 to 109 y). We modelled four profiles representative of Amazonian podzols, two profiles with an old Bh ($^{14}$C age 6.8 $10^3$ and 8.4 $10^3$ y) and two profiles with a very old Bh ($^{14}$C age 23.2 $10^3$ and 25.1 $10^3$ y). The calculated fluxes from the topsoil to the perched water-table indicates that the most waterlogged zones of the podzolized areas are the main source of dissolved organic matter found in the river network. It was necessary to consider two Bh carbon pools to accurately represent the carbon fluxes leaving the Bh as observed in previous studies. We found that the genesis time of the studied soils was necessarily longer than 15 $10^3$ and 130 $10^3$ y for the two younger and the two older Bhs, respectively, and that the genesis time calculated considering the more likely settings runs to around 15 $10^3$ - 25 $10^3$ and 150 $10^3$ - 250 $10^3$ y, respectively.

## 1 Introduction

Podzols are soils characterized by the formation of a sandy, bleached horizon (E horizon) overlying a dark horizon with illuviated organic matter as well as Fe- and Al-compounds (spodic or Bh horizon). In wet tropical areas podzols can be very deep, with E horizons thicker than 10 m and Bh horizons thicker than 4 m (Chauvel et al., 1987; Dubroeucq and Volkoff, 1998; Montes et al., 2011). This means that they can store huge quantities of organic matter: Montes et al. (2011)estimated the C stocks in Amazonian podzols to be around 13.6 Pg C.

This C constitutes a non-negligible portion of the C stored in the Amazonian basin. Indeed, the carbon stored in the aboveground live biomass of intact Amazonian rainforests is estimated to be 93 ± 23Pg C (Malhi et al., 2006). Such large amounts of carbon may play a central role in the global carbon balance (Raymond, 2005), which raises the question of the magnitude of the carbon fluxes during podzol genesis and in response to drier periods that might occur in the future due to climate change. A schematic of the main carbon fluxes in Amazonian podzols (Leenheer, 1980; Lucas et al., 2012; Montes et al., 2011) is presented in Fig. 1. It should be noted that the organic matter (OM) released by the topsoil horizons can be



transferred downwards to the Bh horizons, but may also be rapidly transferred laterally to the river network via a perched water-table on top of the Bh that circulates in the E horizon. The OM stored in the upper part of the Bh can also be

remobilized and be transferred to the river network by the perched water-table. Some of these fluxes have been estimated in a small number of case studies or extrapolated from studies of the chemistry of large rivers (Tardy et al., 2009), but most of them remain unknown. Studies measuring carbon budgets at the profile scale or during soil profile genesis in temperate, boreal or tropical podzols are rare (Schaetzl and Rothstein, 2016; Van Hees et al., 2008). Schwartz (1988) studied giant podzol profiles in the Congo that began to form $40 \cdot 10^3$ y ago but where carbon accumulation in Bh was discontinuous

because of a drier climate between 30 and 12 ky BP. The $^{14}$C age of organic C from the Bh horizon of podzol profiles situated in the Manaus region (Brazil) was found to range from 1960 to 2810 y and it was concluded that the podzols developed in less than $3 \cdot 10^3$ y (Horbe et al., 2004). As pointed out by Sierra et al. (2013), in order to corroborate this conclusion it is necessary to produce a model that accounts for C additions and losses over time. Montes et al. (2011) roughly estimated the C flux to the Bh horizon to be 16.8 gC m$^{-2}$ y$^{-1}$. Sierra et al. (2013) used a compartment model that

was constrained by $^{14}$C dating to estimate the carbon fluxes in a Colombian shallow podzol (Bh upper limit at 0.9 m). They showed that the C fluxes from topsoil horizons to the Bh horizon were smaller (2.1 gC m$^{-2}$ y$^{-1}$) than the fluxes estimated in Montes et al. (2011). However, they did not account for the age and genesis time of the Bh horizon.

In order to better understand the fluxes of C in Amazonian podzols and in particular to determine the rate of carbon accumulation in Bh horizons during podzol genesis, the size of the C fluxes to rivers via both the perched and the deep

water-tables and the vulnerability of the podzol C stocks to potential changes in the moisture regime due to global climate change, data collected from 11 test areas in the high Rio Negro Basin were used to constrain a model of C fluxes (Fig 1). The high Rio Negro basin was chosen because it is a region that has the highest occurrence of podzol in the Amazon (Montes et al., 2011) (Fig. 2). Four representative profiles were selected from a database of 80 podzol profiles to constrain the simulations of C fluxes. We used a system dynamics modelling software package (Vensim) to simulate the formation

of representative Amazonian podzol profiles by constraining both total carbon and radiocarbon with the data collected.

## 2 Methods

### 2.1 Podzol profiles and carbon analysis

Four podzol profiles were selected from our database as representative both from the point of view of the profile characteristics and the $^{14}$C age of the Bh organic matter (Table 1 and Fig. 3). The MAR9 profile was developed on the Icá

sedimentary formation, has a water-logged A horizon, a thin eluvial (E) horizon, a sandy-clay loam Bh with young organic matter (OM) and a low C content; the DPQT profile was developed on a late quaternary continental sediment younger than the Içá formation, has an E horizon of intermediate thickness, a sandy Bh with young OM and a low C content; the UAU4 profile was developed on the Icá sedimentary formation, has an thick E horizon, a sandy Bh with old OM and the C content is high; the P7C profile was developed on crystalline basement rock, has a thick, water-logged O horizon, a E horizon of

intermediate thickness, a silt-loam Bh with old OM and a high C content. It should be noted that in the cases of the DPQT and the UAU4 profiles, the lower limit of the Bh was not reached because of the auger hole collapsed, meaning that for these profiles the Bh C stock is an under-estimate.

Soil samples were analysed for C content with a TOC-LCPN SSM-5000A, Total Organic Carbon Analyzer (Shimadzu). Radiocarbon measurements and conventional age calibration were carried out at the Poznań Radiocarbon Laboratory,

Poland. The conventional age calibration was performed using the program OxCal ver. 4.2 against the INTCAL13





calibration curve. Samples from the topsoils had a pMC (percent Modern Carbon) higher than 100%, which indicates that a significant part of the carbon in the topsoil is post-bomb. We calibrated the age of these samples using a dedicated model described in section 2.2.

The data given in Table 1 were calculated by linear extrapolation of values measured on samples taken at different depths:
between 11 and 28 samples per profile were used for the C stocks calculation and between 6 and 8 samples per profiles were used for radiocarbon measurements.

### 2.2 Model design

We used an approach comparable to previous studies which dealt with carbon budgets and radiocarbon data (e.g. Baisden et al., 2002; Menichetti et al., 2016; Sierra et al., 2013; Tipping et al. , 2012). The model structure, based on the schematic
shown in Fig. 1, and the names of compartments and rate constants are given in Fig. 4. As the turn-over time of the OM in the topsoil horizons is short relative to the average OM turn-over time in the Bh, only one topsoil carbon pool was used, whereas two pools (fast and slow) were used to describe organic carbon dynamics in the Bh horizon. The C can leave the topsoil pool by mineralization, transfer to the Bh pools or to the river by the perched water-table; it can leave the Bh pools by mineralization, transfer to the river by the perched water-table or via the deep water-table. We chose to neglect the flux
of C from the fast Bh pool to the slow Bh pool in order to facilitate the numerical resolution of the system comprising equations describing both the carbon and radiocarbon contents.

The equations describing changes in the carbon content of the different pools are presented below (see Fig. 4 to see the fluxes with which each rate constant is associated):

$$\frac{dC_t}{dt} = C_I - \left(k_t + \alpha_{t-fBh} + \alpha_{t-sBh} + \alpha_{t-r}\right)C_t \tag{1}$$

$$\frac{dC_{fBh}}{dt} = \alpha_{t-fBh}C_{ft} - \left(k_{fBh} + \alpha_{fBh-r} + \alpha_{fBh-d}\right)C_{fBh} \tag{2}$$

$$\frac{dC_{sBh}}{dt} = \alpha_{t-sBh}C_{st} - \left(k_{sBh} + \alpha_{sBh-r} + \alpha_{sBh-d}\right)C_{sBh} \tag{3}$$

where $C_I$ is the C input from litter and roots into the topsoil C pool; $C_t$ the amount of C stored in the topsoil C pool; $C_{fBh}$ and $C_{sBh}$ the amount of C stored in the fast and the slow Bh C pools, respectively; $k_t$, $k_{fBh}$ and $k_{sBh}$ the C mineralization rate constants in the topsoil, the fast Bh and the slow Bh Cpools, respectively; $\alpha_{t-fBh}$ and $\alpha_{t-sBh}$ the transfer rates from the topsoil
pool to the fast and the slow Bh C pools, respectively; $\alpha_{t-r}$, $\alpha_{fBh-r}$ and $\alpha_{sBh-r}$ the transfert rates from respectively the topsoil, the fast Bh and the slow Bh pools to the river by the perched water-table; $\alpha_{fBh-d}$ and $\alpha_{sBh-d}$ the transfer rates from the fast Bh and the slow Bh pools to the deep water-table, respectively.

The equations describing changes in the radiocarbon content of the different pools are the following:


$$\frac{dF_{a\,t}C_t}{dt} = C_I F_{a\,v} - \left(k_t + \alpha_{t-fBh} + \alpha_{t-sBh} + \alpha_{t-r}\right)F_{a\,t}C_t - \lambda F_{a\,t}C_t \tag{4}$$

$$\frac{dF_{a\,fBh}C_{fBh}}{dt} = \alpha_{t-fBh}F_{a\,t}C_t - \left(k_{fBh} + \alpha_{fBh-r} + \alpha_{fBh-d}\right)F_{a\,fBh}C_{fBh} - \lambda F_{a\,fBh}C_{fBh} \tag{5}$$

$$\frac{dF_{a\,sBh}C_{sBh}}{dt} = \alpha_{t-sBh}F_{a\,t}C_t - \left(k_{sBh} + \alpha_{sBh-r} + \alpha_{sBh-d}\right)F_{a\,sBh}C_{sBh} - \lambda F_{a\,sBh}C_{sBh} \tag{6}$$

where $\lambda$ is the [14]C radioactive decay constant, $F_{a\,v}$ the radiocarbon fraction in the organic matter entering the topsoil C pool and $F_{a\,i}$ the radiocarbon fraction in each pool $i$, the radiocarbon fractions being expressed as absolute fraction modern, i.e.
the [14]C/[12]C ratio of the sample normalized for [13]C fractionation to the oxalic acid standard [14]C/[12]C normalized for [13]C fractionation and for radio decay at the year of measurement (Stuiver and Polach, 1977).





With regard to the age calibration of the topsoil organic matter enriched in post-bomb carbon, we considered a single pool that reached a steady state before 1950 (Fig. 5), which allowed the retrocalculation of the radiocarbon fraction $F_{a\,t}$ in 1950 based on the following equation:


$$C_t\,F_{a\,t_{i+1}} = C_t\,F_{a\,t_i} - \lambda\,C_t\,F_{a\,t_i} + \left(F_{a\,v_i} - F_{a\,t_i}\right)C_I \tag{7}$$

where $F_{a\,t_i}$ and $F_{a\,t_{i+1}}$ are the radiocarbon fraction of the topsoil C pool in year $i$ and $i+1$, respectively, and $F_{a\,v_i}$ the radiocarbon fraction in the organic matter entering the topsoil C pool in year $i$. We used the tropospheric $D^{14}CO_2$ record from 1955 to 2011 at Wellington (NIWA, 2016) to estimate the annual value of $F_{a\,v_i}$.

An underlying assumption of this work is that soil formation processes remained constant over time. An alternative

assumption might be, for example, that all the Bh organic matter had accumulated in very short time, after which the Bh was no longer subjected to external exchanges. This scenario could also produce a profile ages close to the observed $^{14}C$ profile ages. Such a case, however, is unlikely. The climate of the high Rio Negro region is likely to have remained humid and forested since the Pliocene, although less humid episodes may have occurred during the Holocene glacial episodes (Colinvaux and De Oliveira, 2001; Van der Hammen and Hooghiemstra, 2000). It is also possible that the rate at which

soil formation proceeded decelerated over time. This will be commented on below.

### 2.3 Model running and tuning

We used the Vensim ® Pro (Ventana Systems inc.) dynamic modelling software to simulate the C dynamics. After setting the initial values for C pools, the model was run in the optimize mode, leaving the model to adjust the rate constants in order to minimise the difference between simulated and measured C pool values and ages. However, frequently the model

did not converge when run in this way. We found that it was because of the great difference between the convergence times between the topsoil C pool and the slow Bh C pool. The long times required to model the genesis of the Bh horizons resulted in numerical errors when modelling the topsoil behavior, because of the values of exponential exponents exceeded the maximum values that the computer could handle (see for example eq. 12 below). To circumvent this technical problem, we optimized the model separately for the topsoils and for the Bh horizons and we found that at the time scale of the

formation of Bh, the topsoil C pool and the topsoil C fluxes to river and Bh horizons could be considered constant.

Although the model structure in Fig. 4 contains two C pools in the Bh horizon, we calculated the numerical solutions of equations considering both carbon budget and radiocarbon age for a single pool Bh in order to determine whether the model could be simplified. Furthermore, this approach allowed us to better assess the weight of the different rate constants in the long-term behaviour of a given pool. The calculation in the simplified configuration is shown in Fig. 6.

In this configuration, the carbon content of the pool is given by:

$$\frac{dC_{Bh}}{dt} = \alpha_{t-Bh}C_t - \beta_{PBh}C_{PBh} \tag{8}$$

where $C_t$ is the amount of C stored in the topsoil pool, $\alpha_{t-sBh}$ the transfer rates from the topsoil pool to the Bh pool, $C_{Bh}$ the amount of C stored in the Bh pool and $\beta_{Bh}$ the transfer rate of C leaving the Bh pool. The solution of this equation with the initial condition $C_{Bh} = C_{0\,Bh}$ when $t = 0$ is:

$$C_{Bh} = \frac{\alpha_{t-Bh}C_t}{\beta_{Bh}} + \left(C_{0\,Bh} - \frac{\alpha_{t-Bh}C_t}{\beta_{Bh}}\right)e^{-\beta_{Bh}t} \tag{9}$$

The equation related to radiocarbon content is the following:

$$\frac{dF_{a\,Bh}C_{Bh}}{dt} = \alpha_{t-Bh}C_t\,F_{a\,t} - (\beta_{Bh} + \lambda)F_{a\,Bh}C_{Bh} \tag{10}$$

where $F_{a\,Bh}$ is the radiocarbon fraction in the Bh.



Considering that the C input from the topsoil to the Bh and its radiocarbon fraction are constant with time, it comes from the two previous equations:

$$\frac{dF_{a\,Bh}}{dt} = \frac{\beta_{Bh}\,\alpha_{t-Bh}C_t\,F_{a\,t} - F_{a\,Bh}\left(\beta_{Bh}\,\alpha_{t-Bh}C_t + \lambda\left(\alpha_{t-Bh}C_t - (\alpha_{t-Bh}C_t - \beta_{Bh}C_{0\,Bh})\,e^{-\beta_{Bh}t}\right)\right)}{\alpha_{t-Bh}C_t - (\alpha_{t-Bh}C_t - \beta_{Bh}C_{0\,Bh})\,e^{-\beta_{Bh}t}} \tag{11}$$

The analytical solution of this equation with the initial condition $F_{a\,Bh} = F_{at}$ when $t = 0$ is:

$$F_{a\,Bh} = \frac{\beta_{Bh}\,F_{a\,t}\,e^{-\lambda t}\left(\beta_{Bh}C_{0\,Bh} + \alpha_{t-Bh}C_t\left(e^{(\beta_{Bh}+\lambda)t} - 1\right) + \lambda\,C_{0\,Bh}\right)}{(\beta_{Bh}+\lambda)\left(\beta_{Bh}C_{0\,Bh} + \alpha_{t-Bh}C_t\left(e^{\beta_{Bh}t} - 1\right)\right)} \tag{12}$$

## 3 Results and discussion

### 3.1 Modelling the formation of a single pool Bh

This section presents conceptual results on the basis of the simplified diagram given on Fig. 6 and in which the flux leaving the Bh is described by a single rate $\beta_{Bh}$. This single rate represents loss from the pool both through the mineralization of

organic carbon, through lateral flow in the perched water-table to the river and through percolation of dissolved organic carbon (DOC) to the deep water-table.

#### 3.1.1 Obtaining the carbon stock

Unsurprisingly, the greater the difference between input and output C fluxes, the faster a given $C_{Bh}$ stock is reached. With a constant input flux and a constant output rate, the output flux progressively increases with time because $C_{Bh}$ increases,

until the input and output fluxes become equal, after which the $C_{Bh}$ reaches a steady state.

When the model is constrained only by the measured values of C stocks, a number of solutions are possible (Fig 7). The example given in Fig. 7 is based on data from the P7C profile (Table 1). Curves 1 and 2 describe the evolution of $C_{Bh}$ with time when the $\beta_{Bh}$ rate is constrained to reach a steady state for the currently observed C stock (158 465 gC m$^{-2}$). The input flux was set at 2.1 g m$^{-2}$ y$^{-1}$ and 16.8 g m$^{-2}$ y$^{-1}$ for curves 1 and 2, respectively, values proposed by Montes et al. (2011) and

Sierra et al. (2013), respectively. The resulting constrained values of $\alpha_{t-Bh}$ and $\beta_{Bh}$ rates are given in the figure. The times required to reach 99% of the steady state values are 345 10$^3$ and 43 10$^3$ y for curve 1 and 2, respectively. The currently observed C stock can be reached in a shorter time, however, if for a given input flux the value of $\beta_{Bh}$ is reduced below the value needed to obtain the currently observed C stock at a steady state. An example is given by the curve 3: the input flux is set at 2.1 g m$^{-2}$ y$^{-1}$, as for curve 1, but the $\beta_{Bh}$ rate is reduced by one order of magnitude. In such a case, it would require

78 10$^3$ y to obtain the currently observed C stock. A value of $\beta_{Bh}$ set to 0 gives the minimum time required to obtain the carbon stock (50 10$^3$ y if the input flux is set to 2.1 g m$^{-2}$ y$^{-1}$).

#### 3.1.2 Obtaining both carbon stock and $^{14}$C age

When the model was constrained by both carbon stock and $^{14}$C age, then a unique solution for reaching the steady state was obtained. This is shown for the P7C profile in Fig. 8 (solid lines), where values of $C_{Bh}$ and $^{14}$C age (158465 gC m$^{-2}$ and

25096 y, respectively), which represent more than 99% of the measured values, were obtained in approximately 590 10$^3$ y; carbon input flux to the Bh and $\beta_{Bh}$ rate were constrained to very low values, 0.95 g m$^2$ y$^1$ and 5.9 10$^{-6}$ y$^{-1}$, respectively. Note that for higher values of the $\beta_{Bh}$ rate, there was no solution because the $^{14}$C age could never be reached.



The simulation of the minimum time required for the observed carbon stock and $^{14}C$ age to be reached is also shown in Fig. 8 (dashed lines). This simulation was obtained by adjusting the input flux with an output flux close to 0 ($\beta_{Bh} = 10^{-10}$).

The minimum time required for the C stock and $^{14}C$ age to be reached and the time required to reach 99% of the C stock and $^{14}C$ age at a steady state are given, along with the associated C input fluxes and $\beta_{Bh}$ rates, in Table 2 for each of the studied profiles. Under each of the conditions, the time required is an exponential function of the $^{14}C$ age of the Bh (Fig. 9).

Taking into account the maximum absolute error propagation does not significantly change the simulation results: the

maximum absolute error on the genesis times is lower than 1,0%, 0.9%, 3,5% and 2,9% for Mar9, DPQT, UAU4 and P7C, respectively. Since such percentages do not alter the orders of magnitude and trends discussed below, the error propagation will not be considered in the following.

The time taken for the Bh horizon of a given profile to form is likely between the two values shown in Table 2 and Fig. 9. The minimum time required for obtaining C stock and $^{14}C$ age is an absolute minimum which assumes that the C output

from the Bh was zero, which is not likely. On the other hand, there is no evidence that a steady state has been reached, especially in the case of the two youngest profiles (MAR9 and DPQT). Consequently, the time taken for the formation of the Bh horizons is very likely comprised between 15 $10^3$ and 65 $10^3$ y for the two youngest profiles and between 140 $10^3$ and 600 $10^3$ y for the two oldest. These results also show that the input C fluxes to the Bh and correspondingly the output C fluxes are 3 to 5 times higher for younger than for older profiles and that the older profiles would have an output rate of

one order of magnitude lower than the younger profiles. It is not immediately clear why such large differences would exist. Previous studies have shown (1) that a part of the accumulated Bh OM is remobilized and exported towards the river network (Bardy et al., 2011); (2) that the water percolating from the Bh to deeper horizons OM contains significant amounts of DOC, even in older profiles (around 2 mg L$^{-1}$, Lucas et al., 2012). These observations are not consistent with very low $\beta_{Bh}$ rates, suggesting that a single Bh C pool is incorrect and that two pools of Bh C are required to adequately represent

Bh C dynamics.

### 3.2 Modelling the formation of the whole profile with a two-pools Bh

#### 3.2.1 Topsoil horizons

As explained in section 2.3., the topsoil horizons were simulated separately because the time needed to reach a steady state is very much shorter for the topsoil horizons than for the Bh horizons. The model outputs for the topsoil horizons of the

studied profiles are given in Table 3.

The results suggest that the topsoil OM in the four profiles needed only between 400 and 700 y to reach a steady state, if the present day topsoils are indeed in a steady-state. The total C flux through the topsoil ($C_t$) is high for the MAR9 profile (286 g m$^{-2}$ y$^{-1}$) and very high for the P7C profile (676 g m$^{-2}$ y$^{-1}$), in accordance with their high topsoil C stock (17722 and 74129 g m$^{-2}$, respectively) and the very young age of their organic matter. Note that the topsoil OM ages are younger than

ages reported by Trumbore (2000) for boreal, temperate or tropical forests. Differences between modelled fluxes through the topsoil are consistent with the field observations: the lowest fluxes (UAU4 and DPQT) correspond to well-drained topsoil horizons, with a relatively thin type Mor A horizons, when the highest fluxes (P7C) corresponds to a podzol having a thick O horizon in a very hydromorphic area. The MAR9 profile is intermediate. It should be noted that the flux through the P7C topsoil would correspond to more than ¾ of the commonly accepted value for the C annually recycled by litter in

equatorial forests (around 850 g m$^{-2}$ y$^{-1}$ – (Wanner, 1970; Cornu et al., 1997; Proctor, 2013).



### 3.2.2 Bh horizons

The partitioning of the C flux leaving the topsoil between the river (rate $\alpha_{t\text{-}r}$), the fast pool of the Bh (rate $\alpha_{t\text{-}fBh}$) and the slow pool of the Bh (rate $\alpha_{t\text{-}sBh}$) is unknown. This is also the case for the partitioning of the C flux from the Bh pools between the river (rates $\alpha_{fBh\text{-}r}$ and $\alpha_{sBh\text{-}r}$) and the deep horizons (rates $\alpha_{fBh\text{-}d}$ and $\alpha_{sBh\text{-}d}$). Consequently, the system is not
sufficiently constrained with the $^{14}$C age of the bulk Bh and there is an infinity of solutions for modelling the Bh formation. We therefore carried out a sensitivity analysis to determine how the main parameters (size of the fast pool of the Bh, C flux input and output C rates for the Bh pools) affected the profile genesis time and to understand the relationships between these parameters.

*Sensitivity to the size of the fast Bh pool* : Fig. 10 shows simulation results with an output C flux from Bh set to be 2 g m$^{-2}$
y$^{-1}$ at end of the genesis time and with values for $C_{fBh}$ ranging from 2.5 10$^3$ to 40 10$^3$ g m$^{-2}$, through 5 10$^3$, 10 10$^3$ and 20 10$^3$. In most configurations, the presence of a fast pool in the Bh extends the time taken for the whole Bh genesis relative to a single-pool Bh. This lengthening of the genesis time increases as a function of the $^{14}$C age of the whole Bh and as a function of the size of the fast Bh pool ($C_{fBh}$).

*Sensitivity to the C fluxes leaving the Bh pools*: the genesis time of the profile lengthens with increasing C flux from the
bulk Bh. The lengthening of the genesis depends, however, on how the C fluxes leaving the Bh C pools vary and on the source of the variation (Fig. 11). In the situation where there is a progressive increase of the Bh output beginning from 0, and this increase is due to the fast Bh pool, the lengthening of the genesis time is fast at first and then slows. An example is given in Fig. 11 for the UAU4 profile for two values of $C_{fBh}$. When the increase is due to the slow Bh pool, the lengthening of the genesis time is slow at first and then becomes very high. An example is given in Fig. 11 for the MAR9 and P7C
profiles, respectively.

The conclusion of this sensitivity study is that, when the size of the fast Bh pool or the C output fluxes from the Bh pools begin to grow from zero, the genesis time of the profiles increases rapidly by a factor of 5 to 20% for the two youngest profiles and 15 to more than 60% for the two oldest profiles.

*Modelling the formation of the whole profiles:* we modelled the formation of the four profiles using the most likely settings
issued from these preliminary results and from the literature. The *C* flux from topsoil to the fast Bh pool was set to be 1 g m$^{-2}$ y$^{-1}$, to get a total C flux from the topsoil to Bh horizons close to the value obtained by Sierra et al. (2013) (2.1 g m$^{-2}$ y$^{-1}$). The $k_t$ mineralization rate was set to 2,57 10$^{-3}$ y$^{-1}$, following preliminary mineralization experiments (unpublished data). The size of the present-day observed fast Bh was set to 5% of the total Bh. As the $k_{fBh}$ mineralization rate had to be set to a value below 1.5 10$^{-4}$ y$^{-1}$ for solutions to be possible, a value of 5 10$^{-5}$ y$^{-1}$ was chosen. The *C* flux from slow Bh to the river
was constrained to a value between 0.1 and 0.2 g m$^{-2}$ y$^{-1}$, to account for the export to river of very humified OM, as observed by Bardy et al. (2011). Results are shown in Fig. 11. The output flux from the whole Bh to deeper horizons was constrained to be between 0,5 and 1 g m$^{-2}$ y$^{-1}$, to account for observations from Montes et al. (2011). Results are shown in Fig. 12.

### 3.3 Age, carbon fluxes and carbon turnover

Considering that the forest litter production is around 850 g m$^{-2}$ y$^{-1}$, the proportion of the litter OM produced by the forest
transferred to the river network is 28, 6, 11 and 57% for profiles MAR9, DPQT, UAU4 and P7C, respectively. This large range of values indicates how waterlogging of the podzol surface horizons affects the transfer of carbon from atmosphere to dissolved organic carbon.

With regard to the Bh horizons, it should be noted that the total C flux leaving these horizons can be distributed in any manner between mineralization, transfer to depth and transfer to the river. However, at least two pools are required for the





total C flux leaving the Bh to be sufficiently large to match the measured values. Obtaining the measured old ages requires a long genesis time (around 195 $10^3$ y for UAU4 and 274 $10^3$ y for P7C) and very small input and output carbon fluxes. Because younger profiles, such as MAR9 and DPQT, can form with higher fluxes, it is likely that the flux rates changed during the development of the profile, reducing progressively with time. Higher flux rates during the earlier periods of profile development, however, would lengthen the profile genesis time (Fig. 11), so that the genesis time estimated here

for the slow Bh (around 17 $10^3$, 22 $10^3$, 195 $10^3$ and 274 $10^3$ for MAR9, DPQT, UAU4 and P7C, respectively) can be considered as a good estimate of the minimum time required to form the presently observed soils. This is especially true for the DPQT and UAU4 profile as their Bh C stock value is a low estimate (cf. §2.1). Such ages are very old when compared to temperate mature podzol that developed in 1 $10^3$-6 $10^3$ y (Sauer et al., 2007; Scharpenseel, 1993).

## 4 Conclusion

Modelling the carbon fluxes by constraining both total carbon and radiocarbon was an effective tool for determining the order of magnitude of the carbon fluxes and the time of genesis of the different carbon-containing horizons. Here modelling the upper horizons separately was necessary because of numerical constrains due to the great differences in carbon turnover time between topsoil horizons and Bh. Steady-state values obtained for the topsoil horizon could subsequently be introduced in Bh modelling. The approach we used can be applied to a wide range of situations, if necessary with

simplifying assumptions to sufficiently reduce the degree of freedom of the system.

The results obtained showed that the organic matter of the podzol topsoil is very young ([14]C age from 62 to 109 y), with an annual C turnover, i.e. the carbon flux passing annually through the horizon, that increases if the topsoil is hydromorphic. This indicates that the most waterlogged zones of the podzolized areas are the main source of dissolved organic matter to the Amazonian hydrographic network.

The model suggests that the Amazonian podzols are accumulating organic C in the Bh horizons at rates ranging from 0.54 and 3.17 gC $m^{-2}$ $y^{-1}$, equivalent to 0.005 to 0.032 tC $ha^{-1}$ $y^{-1}$ of very stable C. Climate models predict changes in precipitation patterns, with greater frequency of dry periods, in the Amazon basin (Meehl and Solomon, 2007), possibly resulting in less frequent waterlogging. The change in precipitation patterns could have a dramatic effect on the C dynamics of these systems with an increase in the mineralisation of topsoil OM and an associated reduction in DOC transfer to both the deep Bh and

the river network.  It may be noted that a [14]C dating of the river DOC would help to determine the proportion of DOC topsoil origin and of Bh horizon origin. The topsoil horizons reached a steady-state in less than 750 y. The organic matter in the Bh horizons was older ([14]C age around 7 ky for the younger profile and 24 $10^3$ y for the older). The study showed that it was necessary to represent the Bh C with two C pools in order to replicate a number of carbon fluxes leaving the Bh horizon that have been observed in previous studies. This suggests that the response of the Bh organic C to changes in

water regime may be quite complex. The formation of the slow Bh pool required small input and output C fluxes (lower than 3.5 and 0.8 g $cm^{-2}$ $y^{-1}$ for the two younger and the two older Bhs, respectively). Their genesis time was necessarily longer than 15 $10^3$ and 130 $10^3$ y for the two younger and the two older Bhs, respectively. The time needed to reach a steady state is very long (more than 48 $10^3$ and 450 $10^3$ y, respectively) so that a steady state was probably not reached. The genesis time calculated by considering the more likely settings runs around 15 $10^3$ - 25 $10^3$ and 180 $10^3$ - 290 $10^3$ y,

respectively; the determination of these ages can help to constrain the dating of the sedimentary formations on which podzols have developed. Finally, a greater frequency of dry periods during the year might also possibly result in an increase in Bh mineralization rates and therefore of $CO_2$ degassing from the Bh, this question will be the object of a further publication.



**Acknowledgments**: This work was funded by grants from (1) Brazilian FAPESP (São Paulo Research Foundation. Process
number: 2011/03250-2; 2012/51469-6) and CNPq, (303478/2011-0; 306674/2014-9), (2) French ARCUS (joint
programme of Région PACA and French Ministry of Foreign Affairs) and (3) French ANR (Agence Nationale de la
Recherche, process number: ANR-12-IS06-0002 "C-PROFOR")

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





**Table 1. The main characteristics of the podzol profiles used in the study. C stock and ages are given ± maximum absolute error.**

| Profile identification | | Depth of E – Bh transition (m) | Topsoil horizons | | Bh | | |
|---|---|---|---|---|---|---|---|
| Name | GPS coordinates | | C stock (gC m$^{-2}$) | Calibrated $^{14}$C age of OM (y) | Texture | C stock (gC m$^{-2}$) | Calibrated $^{14}$C age of OM (y) |
| MAR9 | 00º 49' 48.6'' S 67º 24' 25.1'' W | 0.75 | 17 722 ± 886 | 62 ± 25 | Sandy-clay loam | 55 644 ± 2782 | 6 751 ± 42 |
| DPQT | 00º 15' 24.0' 'N 62º 46' 25.4'' W | 1.6 | 8 056 ± 403 | 108 ± 27 | Sandy | 53 180 ± 2659 | 8 442 ± 37 |
| UAU4 | 00º 10' 11.2'' N 67º 48' 56.3'' W | 6.6 | 7 519 ± 376 | 65 ± 25 | Sandy | 107 813 ± 5391 | 23 193 ± 207 |
| P7C | 00º 36' 42.6'' S 66° 54' 00.6'' W | 1.5 | 74 129 ± 3706 | 109 ± 29 | Silt-loam | 158 465 ± 7923 | 25 096 ± 134 |


**Table 2. Results of simulation for a single pool Bh.**

| | MAR9 | DPQT | UAU4 | P7C |
|---|---|---|---|---|
| Bh $^{14}$C age (y) | 6,751 | 8,442 | 23,193 | 25,096 |
| *Minimum time required for obtaining C stock and $^{14}$C age* | | | | |
| Time (y) | 14,780 | 19,653 | 137,060 | 175,280 |
| Input C flux (gC m$^{-2}$ y$^{-1}$) | 5.52 | 3.68 | 0.84 | 0.95 |
| $\beta_{Bh}$ rate (y$^{-1}$) | 9.8 10$^{-5}$ | 6.9 10$^{-5}$ | 7.5 10$^{-6}$ | 5.9 10$^{-6}$ |
| Current output corresponding to $\beta_{Bh}$ (gC m$^{-2}$ y$^{-1}$) | 5.47 | 3.67 | 0.81 | 0.93 |
| *Time required to reach 99% of the steady state value* | | | | |
| Time (y) | 48,000 | 65,100 | 469,000 | 590,000 |
| Input C flux (gC m$^{-2}$ y$^{-1}$) | 2.6 | 3.6 | 0.77 | 0.89 |
| $\beta_{Bh}$ rate (y$^{-1}$) | 10$^{-10}$ | 10$^{-10}$ | 10$^{-10}$ | 10$^{-10}$ |

**Table 3. Modelling the topsoil horizons. Ct: topsoil C stock; CI: C input flux from roots and litter; Time to steady state: time required to reach 99% of the steady state values for Ct and 14C age; $\beta_t$: sum of the output rates ($\beta_t = k_t + \alpha_{t-r} + \alpha_t - f_{Bh} + \alpha_t$).**

| | MAR 9 | DPQT | UAU4 | P7C |
|---|---|---|---|---|
| $C_t$ (g m$^{-2}$) | 17 722 | 8 056 | 7 519 | 74 129 |
| $^{14}$C age (y) | 62 | 108 | 65 | 109 |
| $C_I$ (g m$^{-2}$ y$^{-2}$) | 286 | 74 | 116 | 676 |
| Time to steady state (y) | 399 | 696 | 420 | 705 |
| $\beta_t$ (y$^{-1}$) | 1.61 10$^{-2}$ | 9.23 10$^{-3}$ | 1.54 10$^{-2}$ | 9.12 10$^{-3}$ |





**Figure caption :**

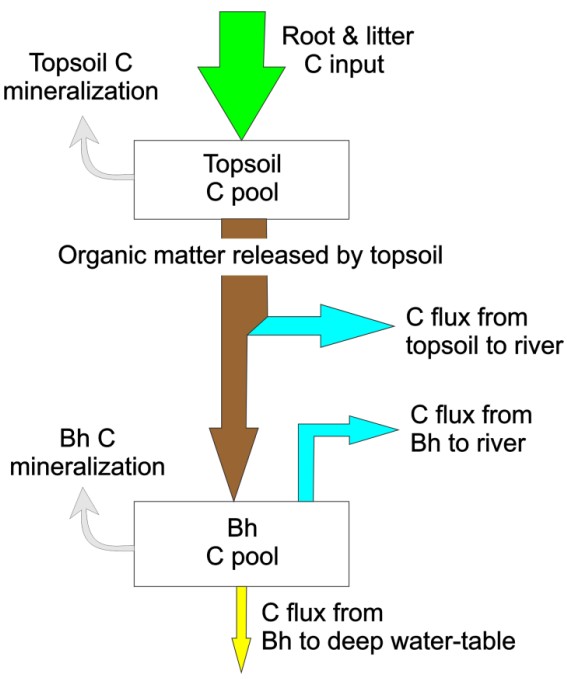

**Figure 1. Schematic of the main C fluxes in a podzol.**

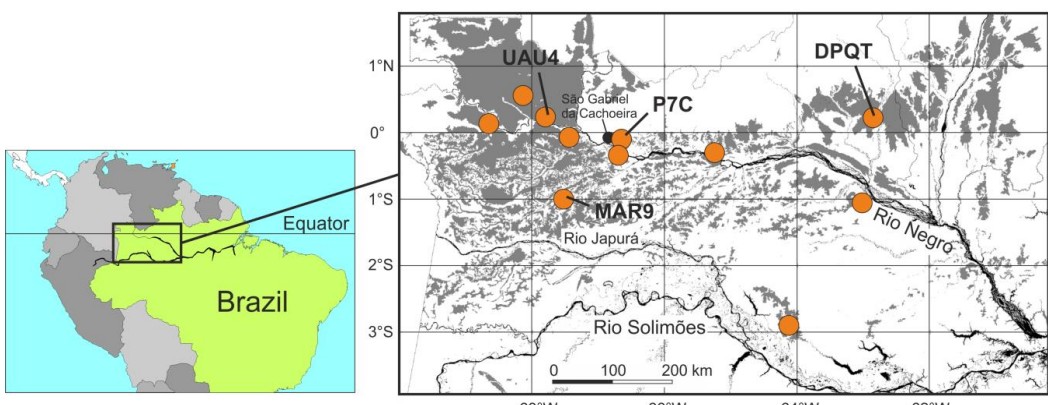

**Figure 2. Location of the studied profiles. Grey areas in the detailed map indicate hydromorphic podzol areas. Orange spots identify test areas.**





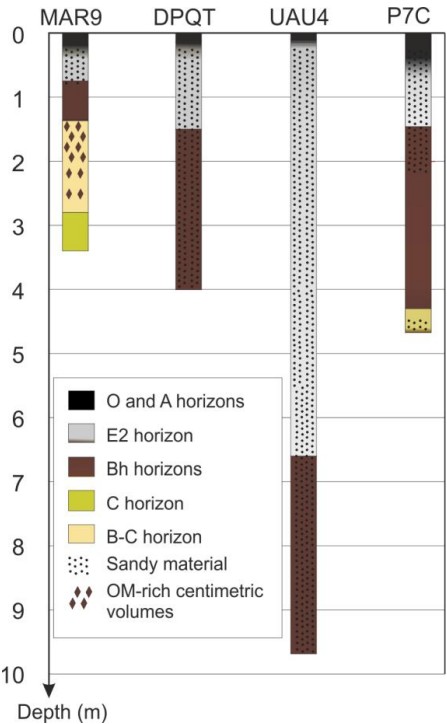


**Figure 3. Sketch of the studied profiles.**




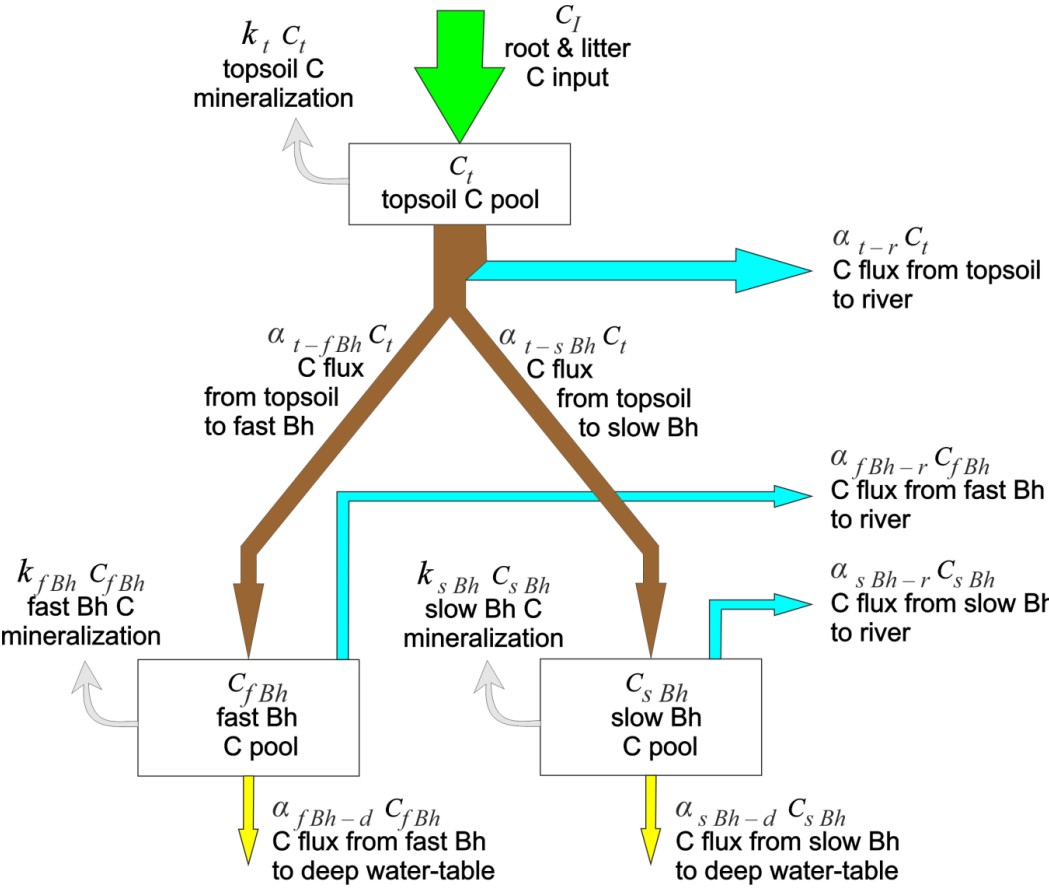

**Figure 4. Model design.**





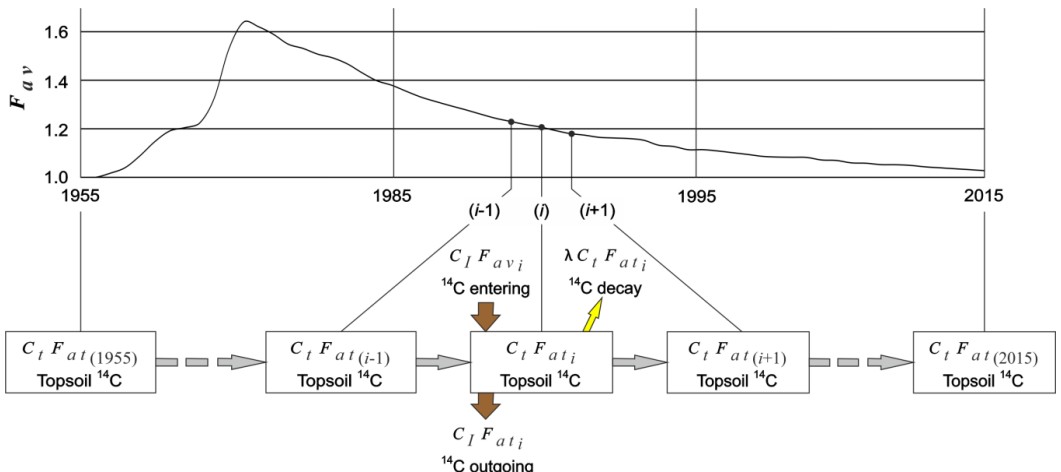

Figure 5. Evolution of the 14C pool in a topsoil that reached a steady state before 1955.

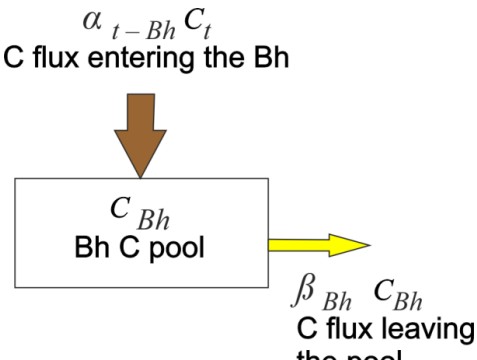

Figure 6. Simplified design for one pool.




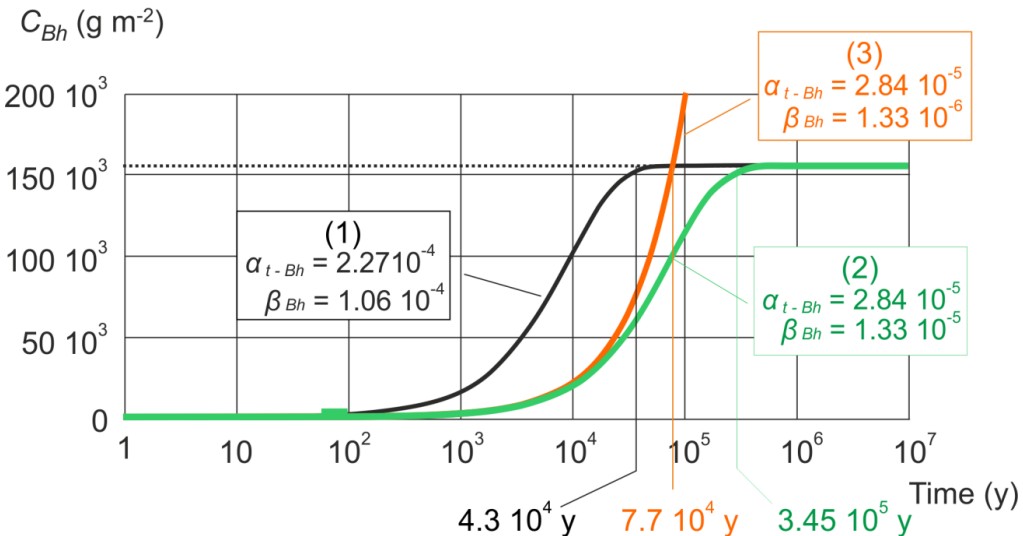


**Figure 7. Single-pool modelling of $C_{Bh}$ of the P7C profile; $C_{0\,Bh}$ set to 0.**


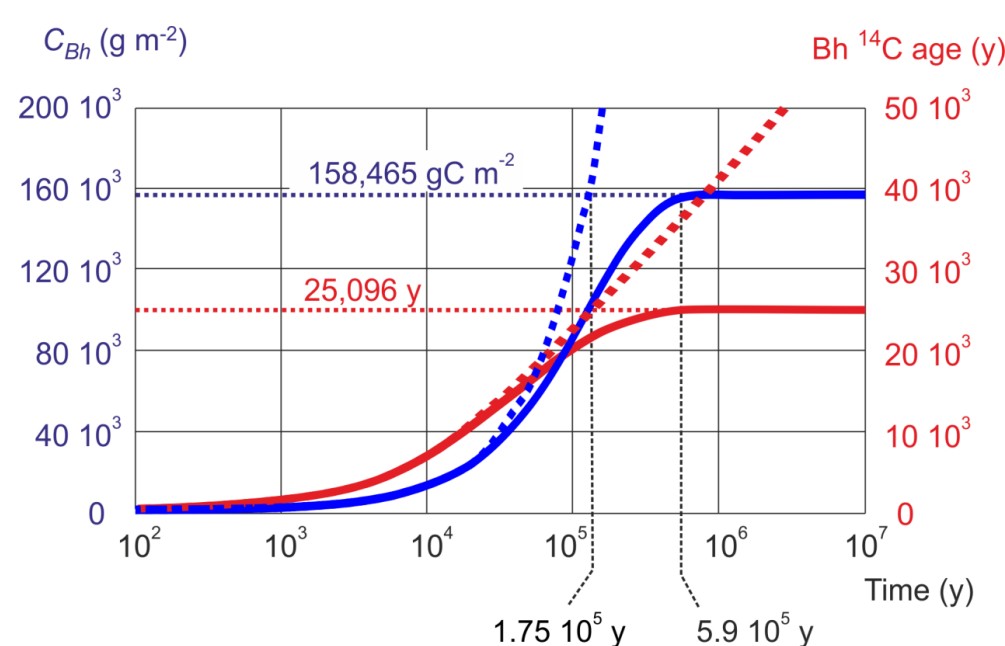

**Figure 8. Single-pool modelling of both CBh and Bh $^{14}$C age of the P7C profile. Corresponding values of C input fluxes and $\beta_{Bh}$ rates are given in Table 2.**





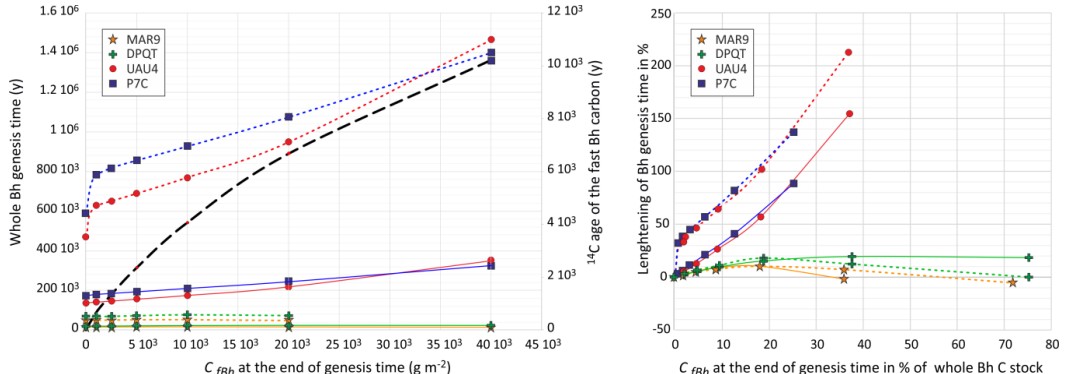

**Figure 9. Relationship between the ¹⁴C age of the Bh and the time needed to form the Bh (single pool modelling).**

**Figure 10. Effect of the fast Bh pool size on the whole Bh genesis time and the ¹⁴C age of the fast Bh. Left graph: absolute values; right graph: values expressed in %.**





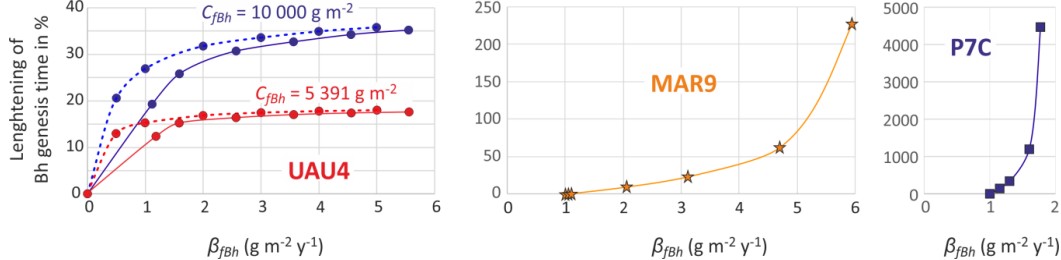

**Figure 11. Effect of constraining the ouput C flux from the Bh on the genesis time. UAU4: effect of the fast Bh output flux. MAR9 and P7C: effect of the slow Bh output flux.**

450

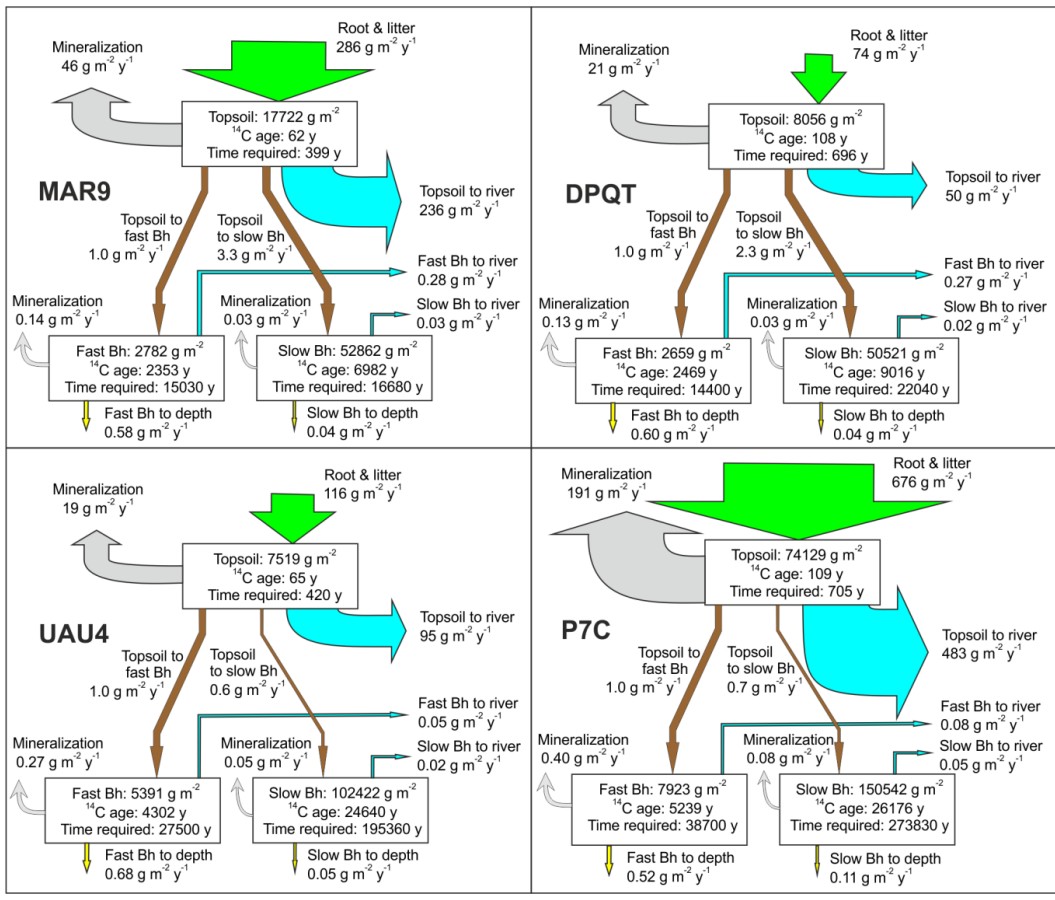

455   **Figure 12. Modelled C fluxes, [14]C ages and C stock in the four studied profiles.**