# Peer review of "Modelling the genesis of equatorial podzols: age and implications for carbon fluxes"

_Biogeosciences, 2016_

## Referee Comment (RC1) · C.A. Sierra (Referee) · 17 Feb 2017

This manuscript presents an analysis on the magnitude of vertical and lateral transfers of carbon in a set of Amazon podzols, as well as estimations of the time required to obtain steady-state values in the Bh horizon. The authors used a simulation model to make these predictions, incorporating not only C stock data but also radiocarbon. This study is important for two main reasons: 1) it helps to clarify previous estimates on the amount of vertical and lateral C transfers for these systems, and 2) it contributes to improve our understanding on the carbon cycle of one of the most important, yet understudied, tropical ecosystems. Despite the importance of the manuscript, I found a number of issues in the model setup and interpretation of radiocarbon data. I will elaborate on these issues below, but I believe that if these issues are adequately addressed, the manuscript will make an important contribution.

[Figure]

My main concern is with the use and interpretation of radiocarbon data. The authors used specific atmospheric radiocarbon curves and standard age calculation procedures to determine the age for the topsoil and Bh horizons. This is a main misinterpretation of the radiocarbon dating method and its application to soils. Standard radiocarbon dating relies on the assumption of a closed system that does not exchange carbon with the surrounding environment, but in soils this assumption does not hold because the system is constantly mixing young and old carbon (Trumbore, 2000). Modeling radiocarbon in soil organic matter usually consists on finding the appropriate value of the decomposition rate that matches both the C stock and the radiocarbon value. The authors have a modeling setup that goes in this direction, but instead of using the radiocarbon data they used an age value to match the decomposition rate. My main concern is in the extra step of finding a $^{14}$C age value and using it for the optimization of the decomposition rates. You do not meet the assumptions for the calculation of a radiocarbon age, however this step is not needed. You can just simply use the radiocarbon data in the $F_a$ notation to match the decomposition rate. For details on the approach see Trumbore et al. (2016); Sierra et al. (2014).

A second concern is related to the arbitrary definitions of the 'minimum time' and the 'genesis time'. For minimum time, the authors use an arbitrary low decomposition constant $\beta$, and for the genesis time an arbitrary proportion of the steady-state value. The problem with these quantities is that they can change dramatically if one changes, for example, $\beta$ from $10^{-10}$ to $10^{-20}$, or the proportion of the steady-state value from 99 to 95%. I understand that these measures are useful to compare the different soils within the context of this analysis, but they may not be very useful to obtain the desired information about the time for soil horizon formation. I would favor instead a less arbitrary definition such as the mean age or the mean transit time (Manzoni et al., 2009).

[Figure]

**Technical comments**

- The $^{14}$C ages presented in the abstract are misleading because you do not meet the closed system assumption. I would rather not present these values, or if you decide to present them, mention that you calculated them even though you do not meet the assumptions of the dating method.

- Line 63. What database? Is it publicly available? Can you provide a reference or a doi?

- Line 75. 'Conventional age calibration' is a contradiction. Conventional radio-carbon age is the age assuming Libby's half life, and does not use a calibration curve. What you probably mean is 'age calibration', but as I mentioned above, this step is not needed for your modeling setup so you may consider eliminating this section from your methods.

- Line 112. I had problems understanding this step and the corresponding Fig 5. You may need to provide additional details.

- Section 2.3. It is not clear from the description of the simulation setup what is the calendar year corresponding to $t = 0$. In other words, did you always started your simulations at a specific calendar year or did this varied for the different soils. This information is important because the atmospheric radiocarbon value corresponding to $t = 0$ influence the forward trajectories for the soil radiocarbon values.

- Equation 8. Why is $P$ a subscript of $\beta$ and $C$? Is this a typo?

- Line 171. These numbers are in reverse order. Curve 1 in Fig 7 has a time required to reach 99% of the steady state of 43 $10^3$, while curve 2, 345 $10^3$. This also makes sense since Montes et al. suggests a much higher value of vertical C

transfers than Sierra et al., therefore the values from Montes et al. should reach the steady-state faster.

- Line 184. This is a very arbitrary definition of minimum time. if $\beta_{Bh}$ is $10^{-20}$, or $10^{-30}$, the 'minimum time' would change drastically, and there is not any relevant reason for why it should be $10^{-10}$. I would recommend not using this concept of minimum time.

- Line 189. What is this maximum absolute error propagation? Did you define this before?

- Lines 247 and 252. Are these decimal numbers? Change comma for point.

- Tables. I'm missing a table with the obtained values of the parameters of the model of Fig 4 obtained for the four different soils. This information is somehow imbedded in Fig 12, but as total stocks and fluxes, and not with the values of the parameters used to obtain these numbers.

- Figures. Figure captions are very poor. Please provide enough information in the caption to better interpret the figures.

**References**

Manzoni, S., Katul, G. G., and Porporato, A. (2009). Analysis of soil carbon transit times and age distributions using network theories. *J. Geophys. Res.*, 114.

Sierra, C. A., Müller, M., and Trumbore, S. E. (2014). Modeling radiocarbon dynamics in soils: Soilr version 1.1. *Geoscientific Model Development*, 7(5):1919–1931.

Trumbore, S. E., Sierra, C. A., and Hicks Pries, C. E. (2016). *Radiocarbon and Climate Change: Mechanisms, Applications and Laboratory Techniques*, chapter Radiocarbon Nomenclature, Theory, Models, and Interpretation: Measuring Age, Determining Cycling Rates, and Tracing Source Pools, pages 45–82. Springer International Publishing.

Trumbore, S. (2000). Age of soil organic matter and soil respiration: radiocarbon constraints on belowground c dynamics. *Ecological Applications*, 10(2):399–411.

---

## Referee Comment (RC2) · Anonymous Referee #2 · 13 Mar 2017

This study used the measurements of carbon stock and 14C of soil carbon at different soil layers to constrain the carbon fluxes into and out of the Bh layer and its carbon turnover rate. Even for a single –pool model of soil carbon in Bh layer at non-steady state, there are three unknowns: influx, efflux and turnover rate, with only two observations for each site (total amount of carbon and carbon age). Theoretically, the optimization problem is under-determined, there will be infinite number of solutions. However only a unique solution was found in this study. Therefore I must have missed additional data constraint used in the optimization by the authors. In general I found that the manuscript provides quite a lot of details and reasoning for the approach taken in this study. However the key message was somehow buried by detail as presented. Significant modifications should be made to distil wealth of information to highlight the key message. That is what are the magnitudes of carbon influx and efflux from the Bh

soil layer and its turnover rate. Estimates of carbon influxes by previous studies varied by one order of magnitude, and result from this study suggests that a lower estimate of the carbon influx is more likely. After presenting the modelling results of the one-pool model for the Bh layer, the section was unfortunately ended with one sentence "These observations are not consistent with very low ïĄćBh rates, suggesting that a single Bh C pool is incorrect and that two pools of Bh C are required to adequately represent Bh C dynamics". I find that quite disappointing. The authors went on to model the formation of the whole profiles with soil carbon Bh being represented using a two-pool model. My question is then how the results in Section 3.1 were used in Section 3.2, ie how the fluxes and turnover rates of the two-pool model for the Bh soil layers are estimated? This is quite unclear to me. In presenting optimization problem, you need to state clearly: observations, optimizing model parameters, the model and optimization method including cost function. This has not been done adequately in this manuscript. Therefore I recommend major revisions. Some additional comments. The results are quite specific to the sites you studied. What are more broad implications? L56 and L58. In L56, you stated that data from 11 test areas were used to constrain a model of C fluxes, but you actually only presented results of four profiles (see L58). Inconsistent! Section 2.3 Would it be simpler to assume that soil carbon pools at different layers were at steady state before 1950 and solve the model analytically at 1950, then integrate the model forward after 1950 to match both the observed carbon pool and age using optimization?
* * *

---

## Author Response (AR1)

**BG-2016-545 Response to referees and marked-up manuscript**

**Response to Referee #1 (Dr. C.A. Sierra)**

**Response to the general comments**

First, we would like to thank Dr. Sierra for his constructive remarks and reassure him about the interpretation of radiocarbon data. We used the $F_a$ radiocarbon data for modelling. In order to compare the Bh between profiles, we calculated a conventional, uncalibrated apparent age from the $F_a$ radiocarbon values. The paragraph from line 86 to 91 is actually confusing: the Poznań Radiocarbon Laboratory has indeed provided both the values of $F_a$ and calibrated ages, the latter having not been used. Regarding the topsoil horizons, the bomb carbon should not be neglected so that we retrocalculated a pre-1950 $F_a$ value that we used for modelling. The paragraph has been modified (lines 90-97), as well some column legends in Table 1 that were incorrect, and we added the reference to Sierra, 2014 because of the good synthesis given in this paper. The $F_a$ values were added in Tables 1 and 2.

Regarding the second concern, and as was pointed out by the reviewer, we used the concept of minimum time and time to steady-state to compare the different soils within the context of this analysis.

- The value we used for the minimum time ($\beta_{Bh} = 10^{-10}$) is not arbitrary: we used a value different from zero for numerical reasons, in order to avoid denominators equal to zero (see for example equ. 12). We checked that the difference between the minimum times obtained using $\beta_{Bh} = 10^{-10}$ and $\beta_{Bh} = 10^{-20}$ is negligible (lower than 0.0005%). We clarified this point in the new version of the manuscript (lines 227-229).
- We agree with the reviewer that the proportion of the steady-state value set to 99% is arbitrary. Values closer to 100% would give a dramatic increase of the time needed to form the profile, as shown on Fig. 8. We used 99% because, as shown on Fig. 8, this value gives a result sufficiently close to the horizontal asymptote to give a reasonable evaluation of the time necessary to reach a steady state. This is now explained on lines 209-211.

**Response to the technical comments.**

Reviewer comments are given in bold, our response in normal font

**• The 14C ages presented in the abstract are misleading because you do not meet the closed system assumption. I would rather not present these values, or if you decide to present them, mention that you calculated them even though you do not meet the assumptions of the dating method.**
We specified in the abstract that the given ages are calculated apparent ages.

**• Line 63. What database? Is it publicly available? Can you provide a reference or a doi?**
This is a database of 80 podzol profiles which have been studied in detail and of which 11 have been dated, this database will be the subject of a further publication. This is now indicated on lines 61-65.

**• Line 75. 'Conventional age calibration' is a contradiction. Conventional radiocarbon age is the age assuming Libby's half life, and does not use a calibration curve. What you probably mean is**

**'age calibration', but as I mentioned above, this step is not needed for your modeling setup so you may consider eliminating this section from your methods.**
This was corrected – see response above.

**• Line 112. I had problems understanding this step and the corresponding Fig 5. You may need to provide additional details.**
We explained better this step: in eq (7), we give the expression of the $F_{a\,t\,i}$ value ($F_a$ value of the topsoil OM on year $i$) as a function of the $F_{a\,t\,i+1}$ value ($F_a$ value of the topsoil OM on year $I$ +1), and we explained on lines 146-148 the iterative retrocalculation.

**• Section 2.3. It is not clear from the description of the simulation setup what is the calendar year corresponding to t = 0. In other words, did you always started your simulations at a specific calendar year or did this varied for the different soils. This information is important because the atmospheric radiocarbon value corresponding to t = 0 influence the forward trajectories for the soil radiocarbon values.**
The simulations did not started at a specific year. We used the present day atmospheric carbon value to simulate the Bh formation. This is an approximation, as it is known that atmospheric radiocarbon value was higher than at present (see for example Kitagawa and Van der Plicht, 1998; Reimer et al., 2009), which likely leads to a systematic underestimation of development durations, although this underestimation remains low compared to uncertainty. We addressed this question in section 3.3, lines 349-352.

**• Equation 8. Why is P a subscript of _ and C? Is this a typo?**
It was a "copy and paste" error and was corrected.

**• Line 171. These numbers are in reverse order. Curve 1 in Fig 7 has a time required to reach 99% of the steady state of 43 103, while curve 2, 345 103. This also makes sense since Montes et al. suggests a much higher value of vertical C C3 transfers than Sierra et al., therefore the values from Montes et al. should reach the steady-state faster.**
OK, this was corrected.

**• Line 184. This is a very arbitrary definition of minimum time. if _Bh is 10−20, or 10−30, the 'minimum time' would change drastically, and there is not any relevant reason for why it should be 10−10. I would recommend not using this concept of minimum time.**
See explanation above – this minimum time is really significant, we used $10^{-10}$ in place of 0 for numerical reasons but the result is the same as if 0 is used. We also checked this using equations rewritten for $\beta_{Bh} = 0$.

**• Line 189. What is this maximum absolute error propagation? Did you define this before?**
This is the maximum absolute error, this was corrected in the text (line 241).

**• Lines 247 and 252. Are these decimal numbers? Change comma for point.**
This was corrected.

**• Tables. I'm missing a table with the obtained values of the parameters of the model of Fig 4 obtained for the four different soils. This information is somehow imbedded in Fig 12, but as total stocks and fluxes, and not with the values of the parameters used to obtain these numbers.**
We added a table to give the values of the parameters (Table 4).

**• Figures. Figure captions are very poor. Please provide enough information in the caption to better interpret the figures.**
This problem corresponded to a poor framing of the figures, which cut some of the legend given within the figures. It was corrected in the new version.

**Response to Referee #2**

The referee comments are given in quotation marks. As a preamble, we emphasize that most of the answers to the referee's questions can be found in the original manuscript. We agree, however, with the referee that several explanations were unclear and we have improved the manuscript in this regard.

*Response to the general comments*

*"This study used the measurements of carbon stock and 14C of soil carbon at different soil layers to constrain the carbon fluxes into and out of the Bh layer and its carbon turnover rate. Even for a single –pool model of soil carbon in Bh layer at non-steady state, there are three unknowns: influx, efflux and turnover rate, with only two observations for each site (total amount of carbon and carbon age). Theoretically, the optimization problem is under-determined, there will be infinite number of solutions. However only a unique solution was found in this study. Therefore I must have missed additional data constraint used in the optimization by the authors."*

Response: In a first step (§ 3.1.2) we used the single-pool model to determine two particular solutions for which there are only two unknowns, the conditions for obtaining the stationary state (input = output) and the minimum formation time (output = 0). This first step showed that the output fluxes from older Bhs were too low compared to the measures reported by previous studies and that it is therefore necessary to consider two Bh pools. In the second step (§ 3.2.2) we clearly stated that " there is an infinity of solutions for modelling the Bh formation" (line 225 of initial manuscript) and that "We therefore carried out a sensitivity analysis to determine how the main parameters (size of the fast pool of the Bh, C flux input and output C rates for the Bh pools) affected the profile genesis time and to understand the relationships between these parameters" (lines 226-228 of initial manusript). This analysis and data from literature allowed to exclude unrealistic values and to constrain C fluxes and mineralization rates, as explained on lines 245 to 251 of old manuscript. To better explain the data constraint we modified the manuscript on lines 263-266 and lines 316-326 of the new version of the manuscript.

*"In general I found that the manuscript provides quite a lot of details and reasoning for the approach taken in this study. However the key message was somehow buried by detail as presented. Significant modifications should be made to distil wealth of information to highlight the key message. That is what are the magnitudes of carbon influx and efflux from the Bh soil layer and its turnover rate."*

Response: The magnitude of carbon influx and efflux to the Bh pools are given on Fig. 12. We did not specify the turn-over rates because the slow pool of the Bh is not in a steady state. However, we agree with the referee that it would be informative to give instantaneous turnover times at present day, so we added these values on lines 318-319.

*"Estimates of carbon influxes by previous studies varied by one order of magnitude, and result from this study suggests that a lower estimate of the carbon influx is more likely."*

Response: It is indeed our conclusion, which is even more important for us that several authors of the present study were also authors of the previous study which gave a higher estimate.

*"After presenting the modelling results of the one-pool model for the Bh layer, the section was unfortunately ended with one sentence "These observations are not consistent with very low A to Bh rates, suggesting that a single Bh C pool is incorrect and that two pools of Bh C are required to adequately represent Bh C dynamics". I find that quite disappointing."*

Response: The ending sentence cited by the referee was preceded by another sentence (lines 201-203 of old manuscript, ), which, based on data from the literature, explained that the Bh output flow should be at least 2 mgC L-1, which is not consistent with the 1-pool model. We think we need to

demonstrate to the reader that the one pool model was not suitable. For more clarity, we changed the text in the new version (lines 257-259).

*"The authors went on to model the formation of the whole profiles with soil carbon Bh being represented using a two-pool model. My question is then how the results in Section 3.1 were used in Section 3.2, ie how the fluxes and turnover rates of the two-pool model for the Bh soil layers are estimated? This is quite unclear to me. In presenting optimization problem, you need to state clearly: observations, optimizing model parameters, the model and optimization method including cost function. This has not been done adequately in this manuscript. Therefore I recommend major revisions."*

Response: Section 3.1 was necessary to show, from a simplified system, the conditions for obtaining the stationary state and the minimum formation time. These notions seem indispensable to us to understand the sensitivity study given in § 3.2.2. As explained above, to improve the presentation of the problem, we better explained at the end of section 3.1 why the one pool model is incorrect (lines 257-259) and, agreeing with the referee that observations, optimizing model parameters and cost functions were not clearly presented, we amended the text accordingly: lines 263 to 266 for the topsoil modelling, lines 316-326 for the Bh modelling.

*"Some additional comments.*

*The results are quite specific to the sites you studied. What are more broad implications? L56 and L58. In L56, you stated that data from 11 test areas were used to constrain a model of C fluxes, but you actually only presented results of four profiles (see L58). Inconsistent!""*

Response: We agree with the referee (as well with referee #1) that this is unclear. For more clarity, we explained now on lines 61-65 that four podzol profiles were selected from a database of 80 podzol profiles (from 11 test areas) which have been studied in detail and of which 11 have been dated. This database will be the subject of a further publication dedicated to the podzolic system genesis. These profiles were selected as representative both from the point of view of the profile characteristics and the 14C apparent age of the Bh organic matter stock (lines 73-74).

*"Section 2.3 Would it be simpler to assume that soil carbon pools at different layers were at steady state before 1950 and solve the model analytically at 1950, then integrate the model forward after 1950 to match both the observed carbon pool and age using optimization?"*

Response: As stated in section 3.1.2, there is no evidence that a steady state has been reached, especially in the case of the two youngest profiles. Considering the possible C influx values in the Bh, which are very small with regard to the C stock, no correction for the bomb carbon was needed. This is not true for the topsoil horizons, which show Fa values greater than 1: a correction for bomb carbon was therefore necessary to calculate the steady state conditions and the Fa values to be used for the carbon transferred for topsoil to the Bh pools.

[revised manuscript text omitted]

horizons was constrained to be between 0,5 and 1 g m$^{-2}$ y$^{-1}$, to account for observations from Montes et al. (2011). Results are shown in Fig. 12As the $k_{fBh}$ and the $k_{sBh}$ mineralization rate had to be set below 1 10$^{-4}$ and 1 10$^{-6}$ y$^{-1}$, respectively, for solutions to be possible, values of 5 10$^{-5}$ and 5 10$^{-7}$ y$^{-1}$, respectively, were chosen. Optimizing parameters were $\alpha_{1\text{-}sBh}$, $\beta_{fBh}$ and $\beta_{sBh}$ and a multiple cost function was minimizing the differences between modelled and observed value for $C_{Bh}$ and $F_{a\,Bh}$. Results are shown in Fig. 12 and corresponding parameters in Table 4. The resulting present day instantaneous turnover times of C in the whole Bh are 12940, 16115, 67383 and 98215 gC for profiles MAR9, DPQT, UAU4 and P7C, respectively.

[Figure]

[Figure]

360

**Figure 12. Modelled C fluxes, ¹⁴C ages and C stock in the four studied profiles.**

**Table 4. Parameters used for the modelling shown in Fig. 12.**

[revised manuscript text omitted]

